# On the Theoretical Analysis of Dense Contrastive Learning

## Abstract

Contrastive learning has achieved outstanding performance in self-supervised learning. However, the canonical image-level matching pretext is unsuitable for multi-object dense prediction tasks like segmentation and detection. Recently, numerous studies have focused on dense contrastive learning (DCL) that adopts patch-level contrast to learning representations aware of local information. Although empirical evidence has validated its superiority, to date, there has not been any theoretical work that could formally explain and guarantee the effectiveness of DCL methods, which hinders their principled development. To bridge this gap, using the language of spectral graph theory, we establish the first theoretical framework for modeling and analyzing DCL by dissecting the corresponding patch-level positive-pair graph. Specifically, by decoupling the image-level and patch-level supervision, we theoretically characterize how different positive pair selection strategies affect the performance of DCL, and verify these insights on both synthetic and real-world datasets. Furthermore, drawing inspiration from the theory, we design two unsupervised metrics to guide the selection of positive pairs.

## 1 Introduction

In recent years, contrastive learning in computer vision has gained attention for its success in image classification, rivaling supervised learning methods (Caron et al., 2020; 2021; Chen et al., 2020; Chen & He, 2021; Grill et al., 2020; He et al., 2020). However, most of these works primarily focus on image classification as the downstream task, while the transferability of contrastive learning methods to dense prediction tasks such as object detection and semantic segmentation has received less attention. Notably, the key distinction between these tasks lies in the fact that image classification only requires global image features, whereas dense prediction tasks necessitate representations that encompass pixel-level information. Traditional contrastive learning methods typically learn global representations as their pretext tasks are designed at the image level, which limits their performance on dense prediction tasks. This limitation prompted the emergence of approaches that specifically target the learning of local features within the pretext task (Wang et al., 2021; Xie et al., 2021; Ding et al., 2022; Wei et al., 2021; Zhou et al., 2021; Zhang et al., 2022; 2023; Yun et al., 2022). These methods, collectively referred to as dense contrastive learning (DCL), perform contrastive learning on the image patches[1], thereby enabling the learning of pixel-level representations.

Although DCL has developed rapidly in terms of model design, current mainstream theoretical work on contrastive learning primarily concentrates on global contrastive learning (GCL), which refers to traditional image-level contrastive learning (Arora et al., 2019; HaoChen et al., 2021; Tsai et al., 2021; Wang & Isola, 2020; Wang et al., 2022a;b). Consequently, there is still a lack of relevant analysis on the working principles and generalization ability of DCL. While theories from GCL can be applied to DCL by simply transferring the analysis unit from images to patches, they do not offer additional constructive insights such as guidance for selecting positive pairs. This limits our understanding of the unique characteristics and effectiveness of DCL. To the best of our knowledge, there is only one work that analyzes DCL from a theoretical perspective (Moon et al., 2022). However, the theoretical part of this work simply applies the alignment-uniformity theory (Arora et al., 2019) from GCL to DCL and does not provide further insights on the generalization ability or model design of DCL.

---

[1]Here, the term "patches" encompasses both pixels in the feature maps of ResNets (He et al., 2016) and patches in ViTs (Dosovitskiy et al., 2020). We use the term "patches" collectively, as pixels in feature maps also represent the features of specific regions in the original views.

When examining the mechanism of DCL, it becomes apparent that in GCL, positive pairs are selected based solely on image-level supervision. However, in DCL, both image-level and patch-level supervision are considered, and the unique feature of DCL lies in the use of local information to form positive pairs, which varies across different methods. As found in many DCL works, the strategy of selecting positive samples has a dramatic influence on downstream performance (Xie et al., 2021; Yun et al., 2022). Given that most strategies rely solely on patch-level supervision, it is critical to decouple the two types of supervision for matching positive pairs in the theoretical analysis of DCL.

To address the aforementioned issues, we first formulate the positive pairs using a patch-level positive-pair graph. By decomposing the adjacency matrix of this graph using the Kronecker product, we decouple the two kinds of supervision into two adjacency matrices. Our analysis of the matrix corresponding to patch-level supervision sheds light on how various strategies for selecting positive pairs impact the DCL model performance. Moreover, our theory inspires new unsupervised metrics to evaluate the strategy of positive pair selection. We summarize the main contributions as follows.

- By constructing a patch-level positive-pair graph on the dataset, we establish a theoretical framework for DCL, and for the first time give a theoretical guarantee on the downstream performance of DCL.
- Using the Kronecker product to decompose the adjacency matrix of the patch-level positive-pair graph, we are able to decouple the image-level and patch-level supervision that are utilized in the positive pair sampling process. By analyzing the properties of the matrix corresponding to the patch-level supervision, we theoretically find a trade-off between the quantity and correctness of positive pairs.
- To empirically validate the proposed understanding, we construct a new dense-labeled synthetic dataset using the Gaussian mixture model. We then conduct DCL experiments on this dataset, and the results effectively demonstrate the trade-off that aligns with our theory.
- Motivated by the theory, we introduce two unsupervised metrics that serve as guidelines for selecting positive pairs in DCL, which are strongly correlated with model performance.

## 2 THEORETICAL ANALYSIS FOR DENSE CONTRASTIVE LEARNING

### 2.1 PRELIMINARY: SPECTRAL CONTRASTIVE LEARNING

We begin by introducing the framework of spectral contrastive learning, which was proposed in HaoChen et al. (2021) for GCL. Let $\overline{\mathcal{X}}$ denote the set of all natural images. We assume that each image belongs to one of $r$ classes, and let $y : \overline{\mathcal{X}} \to [r]$ denote the ground-truth labeling function. For a given natural image $\bar{x} \in \overline{\mathcal{X}}$, we use $\mathcal{A}(\cdot|\bar{x})$ to denote the distribution of its augmentations. The set of all augmented views, denoted by $\mathcal{X}$, has a size $n = |\mathcal{X}|$ that is assumed to be exponentially large but finite. The distribution of $\overline{\mathcal{X}}$ is denoted as $\mathcal{P}_{\overline{\mathcal{X}}}$. Then, we define an image-level positive pair graph $G_I(\mathcal{X}, w^I)$ to represent the relationships between positive pairs. Its vertices are the augmented views in $\mathcal{X}$. The edges are defined by the joint probability of two augmented views being sampled as a positive pair, given by $w^I_{xx'} = \mathbb{E}_{\bar{x} \sim \mathcal{P}_{\overline{\mathcal{X}}}}[\mathcal{A}(x|\bar{x})\mathcal{A}(x'|\bar{x})]$. We denote its adjacency matrix as $A_I := (w^I_{xx'})_{x,x' \in \mathcal{X}} \in \mathbb{R}^{n \times n}$. Denote the normalized adjacency matrix as $\bar{A}_I := D^{-1/2} A_I D^{-1/2}$, where $D := diag(w^I_x)_{x \in \mathcal{X}}$, $w^I_x := \sum_{x' \in \mathcal{X}} w^I_{xx'}$. Let $\lambda_1, \cdots, \lambda_k$ be the $k$ largest eigenvalues of $\bar{A}_I$.

It can be proved that training a neural network $f_I : \mathcal{X} \to \mathbb{R}^k$ by optimizing the spectral contrastive loss, as defined below, is equivalent to performing spectral clustering on the positive-pair graph $G_I$.

$$\mathcal{L}(f_I) := -2 \cdot \mathbb{E}_{x,x^+} \left[ f_I(x)^\top f_I(x^+) \right] + \mathbb{E}_{x,x'} \left[ \left( f_I(x)^\top f_I(x') \right)^2 \right]. \tag{1}$$

Proof of the equivalence between optimizing equation 1 and spectral clustering is given in Appendix D. Here, $(x, x^+)$ is drawn with probability $w^I_{xx^+}$, which is the positive pair, while $x$ and $x'$ are i.i.d. drawn from $\mathcal{X}$, which is the negative pair. Thus, if two augmented views are connected in the image-level positive-pair graph $G_I$, they are considered as a positive pair in spectral contrastive learning. In order to give the theoretical guarantee for GCL, we make the following assumptions.

**Assumption 2.1** (Realizability). *Let $\mathcal{F}$ be a hypothesis class containing functions from $\mathcal{X}$ to $\mathbb{R}^k$. We assume that at least one of the global minima of $\mathcal{L}(f_I)$ belongs to $\mathcal{F}$.*

**Assumption 2.2** (Labeling error). *Consider a natural data sample $\bar{x} \sim \mathcal{P}_{\overline{\mathcal{X}}}$ and let $y_I(\bar{x})$ denote its label. Let the augmentation $x \sim \mathcal{A}(\cdot|\bar{x})$. We assume the existence of a classifier $g$ that can predict $y_I(\bar{x})$ given $x$ with error at most $\varepsilon$. In other words, $g(x) = y_I(\bar{x})$ with probability at least $1 - \varepsilon$.*

Assumption 2.1 ensures that the function class of neural networks is sufficiently large. Assumption 2.2 assumes that augmentations can rarely convert the labels of the augmented views, i.e., labels are recoverable from augmentations. Then, we can derive the following theoretical guarantee for GCL.

**Theorem 2.3.** *Denote $\mathcal{E}(f_I)$ as the linear probe error, i.e., the error of the best possible linear classifier on the representations calculated by the encoder $f_I$. We can upper-bound it as*

$$\mathcal{E}(f_I) \lesssim \frac{2\varepsilon}{1 - \lambda_{k+1}}. \tag{2}$$

*where $\lambda_{k+1}$ is the $(k+1)$-th largest eigenvalue of $\bar{A}_I$.*

The formal definition of $\mathcal{E}(f_I)$ is given in the Appendix C.1.

## 2.2 A General Theoretical Framework for Dense Contrastive Learning

In the context of DCL, the key distinction from GCL is the utilization of patch-level representations for calculating the contrastive loss. We notice that the positive-pair graph framework in Section 2.1 can naturally incorporate this difference by considering the patch as the node of the positive-pair graph. Denote $\overline{\mathcal{X}}_P$ as the set of patches in the natural images, namely the natural patches. The natural patches are then augmented into augmented patches (according to the data augmentation performed on the images), which form the set $\mathcal{X}_P$. In particular, we assume that each patch has a unique label $y(p)$ assigned to it, where $y : \mathcal{X}_P \to [r]$ and $r$ denotes the number of ground-truth classes.

Then, we define a patch-level positive-pair graph $G(\mathcal{X}_P, w)$ to denote the relationships between the patches. Similar to the image-level scenario, the edge weight of the adjacency matrix $A$ is defined as

$$w_{pp'} := \mathbb{E}_{\bar{p} \sim \mathcal{P}_{\overline{\mathcal{X}}_P}} [\mathcal{A}(p|\bar{p})\mathcal{A}(p'|\bar{p})]. \tag{3}$$

Now we can derive the theoretical guarantee for DCL.

**Theorem 2.4.** *Define the labeling error rate $\alpha$ as the average label disagreement among the positive pairs $(p, p^+)$, i.e., $\alpha := \mathbb{E}_{p,p^+} \mathbb{1}[y(p) \neq y(p^+)]$. Let $\mathcal{E}(f)$ be the linear probe error for DCL, where $f : \mathcal{X}_P \to \mathbb{R}^k$ is the encoder function on patches. We can upper-bound it as*

$$\mathcal{E}(f) \lesssim \frac{\alpha}{1 - \gamma_{k+1}}. \tag{4}$$

*where $\gamma_{k+1}$ is the $(k+1)$-th largest eigenvalue of $\bar{A}$.*

This theorem provides an error bound for the task of predicting the labels of patches. Note that the denominator of the error bound becomes the labeling error rate $\alpha$, but not $2\varepsilon$ in equation 2. This distinction arises from our consideration of the DCL scenario, where patch augmentations, particularly cropping, are likely to alter the class of the patches. Consequently, Assumption 2.2 does not hold for a sufficiently small value of $\varepsilon$.

## 2.3 Decoupling Image-level and Patch-level Supervision using Kronecker Product

We observe that while the natural definition of edge weights (Equation 3) provides a mathematical characterization of DCL, it does not offer additional insights compared to the results in GCL. Specifically, DCL methods primarily diverge in their strategies for selecting positive pairs, which remains independent of the data augmentations. Unfortunately, the modeling in the last section struggles to characterize the strategy effectively, as the edge weight defined in Equation 3 is solely dependent on data augmentation. This limitation hinders our ability to analyze the essence of DCL and provide unique insights distinct from the GCL scenario. Therefore, we propose a refined definition for the patch-level positive-pair graph that decouples these two types of supervision (data augmentation and positive pair selecting strategy). As discussed in Section 2.4, this alternative modeling of the patch-level positive-pair graph allows us to analyze how different strategies for selecting positive pairs impact the performance of DCL.

First, we define the process of positive patch pair sampling, as showed in Figure 1. The process begins by transforming a natural image into augmented views. In the overlapping region of the augmented views, we select several patches from each view with exact positional correspondence on the natural image (depicted by red boxes). This is equivalent to regarding the image-level augmentation as a

composition of the cropping operator and the taking-overlap operator. Following this, the positive pair selection strategy aligns patches that demonstrate similarity in terms of features or geometry.

It is crucial to note that the requirement for patches to be chosen in the overlapping area of the augmented views is significant; otherwise, the correspondence between patches from different augmented views may be compromised. In practice, DCL methods often adhere to this principle, typically first finding the common area of two augmented views and only performing dense alignment within this area (Ding et al., 2022; Xie et al., 2021). Assume that we choose $m$ patches from each augmented view, we give the definition of the edge weight as follows:

$$w_{p_i p'_j} = \mathbb{E}_{\bar{x} \sim \mathcal{P}_{\overline{X}}}[\mathcal{A}(x|\bar{x})\mathcal{A}(x'|\bar{x})] \cdot B_{ij}. \tag{5}$$

In this equation, $x$ and $x'$ respectively denote the views to which random patches $p$ and $p'$ belong, and $B \in \{0,1\}^{m \times m}$ is a symmetric Boolean matrix. The first term characterizes the probability of two augmented views being generated from the same natural image, capturing the supervision of data augmentation. The second term serves as an indicator function, governing the positive pair selection strategy inside a single view. It represents the supervision of the strategy. Here, we name $B$ as the *strategy matrix*. Specifically, we assume that when $x$ and $x'$ form an image-level positive pair, $p_i$ and $p'_i$ correspond to patches that are mapped from the same region of the original image.

This definition implicitly assumes that the strategy for matching positive pairs within a view is consistent across all views. This assumption is reasonable since we have restricted the patch selection in the overlapping region of the augmented views. In fact, works like Ding et al. (2022); Wei et al. (2021) indeed use a fixed alignment rule (i.e., a universal $B$) among patches. Furthermore, We can simplify the expression of the adjacency matrix using the Kronecker product:

$$A = A_I \otimes B. \tag{6}$$

In the form of this equation, the $(i+jm)$-th vertex is the $i$-th patch in the $j$-th augmented view. Under this formulation, the constructed graph exhibits a hierarchical structure, as illustrated in Figure 1.

It is clear that the image-level adjacency matrix $A_I$ represents the supervision of data augmentation, corresponding to the image-level similarity of patches, while the strategy matrix $B$ represents the supervision of positive pair selecting strategy, corresponding to the patch-level similarity. Equation 6 utilizes the Kronecker product to decomposite the patch-level adjacency matrix $A$, effectively decoupling the two types of supervision. The Kronecker product possesses favorable eigen-properties, allowing us to analyze the impact of various positive pair selection strategies on model performance based on the generalization error bound presented in the previous section (Theorem 2.4). Specifically, Lemma 2.5 shows that it is convenient to calculate the normalized adjacency matrix of the Kronecker product of two adjacency matrices. Lemma 2.6 gives the eigen-properties of Kronecker product.

**Lemma 2.5.** *Let $\bar{(\cdot)}$ be the normalized form of a matrix defined in Section 2.1. Given $A = A_I \otimes B$, we have*

$$\bar{A} = \bar{A}_I \otimes \bar{B}. \tag{7}$$

**Lemma 2.6.** *Let $A_1 \in \mathbb{R}^{n \times n}$ have eigenvalues $\lambda_i, i \in [n]$, and let $A_2 \in \mathbb{R}^{m \times m}$ have eigenvalues $\mu_j, j \in [m]$. Then the $mn$ eigenvalues of $A_1 \otimes A_2$ are $\lambda_1 \mu_1, \ldots, \lambda_1 \mu_m, \lambda_2 \mu_1, \ldots, \lambda_2 \mu_m, \ldots, \lambda_n \mu_m$.*

Let the eigenvalues of $\bar{A}_I$ be $\lambda_1, \lambda_2, \cdots, \lambda_n$, and the eigenvalues of $\bar{B}$ be $\mu_1, \mu_2, \cdots, \mu_m$, which are ordered from the largest to the smallest. By combining Lemma 2.5 and 2.6, the eigenvalues of $\bar{A}$ are given by $\lambda_1 \mu_1, \ldots, \lambda_1 \mu_m, \lambda_2 \mu_1, \ldots, \lambda_2 \mu_m, \ldots, \lambda_n \mu_m$. Thus, if we assume that both the dataset and the class of data augmentation as consistent (i.e., $A_I$ is consistent), the eigenvalues of the normalized patch-level adjacency matrix $\bar{A}$, which are $\{\gamma_i\}$, are positively correlated with the eigenvalues of the normalized strategy matrix $\bar{B}$, which are $\{\mu_j\}$. In this scenario, by analyzing the distribution of $\{\mu_j\}$, we can gain insights into how different positive pair selection strategies affect the generalization ability of DCL models.

## 2.4 THEORETICAL CHARACTERIZATION OF POSITIVE PAIR SELECTION

After decoupling the two types of supervision in the modeling, we can analyze how different strategies for selecting positive pairs impact the generalization ability of DCL. To ensure that the positive pair selection strategy is the sole factor affecting model performance, in the following, we assume that the image-level adjacency matrix $A_I$ is consistent.

From equation 4 we know that the performance of DCL is influenced by two main factors: the error rate $\alpha$ and the eigenvalues of the normalized patch-level adjacency matrix $\bar{A}$. Given our assumption of

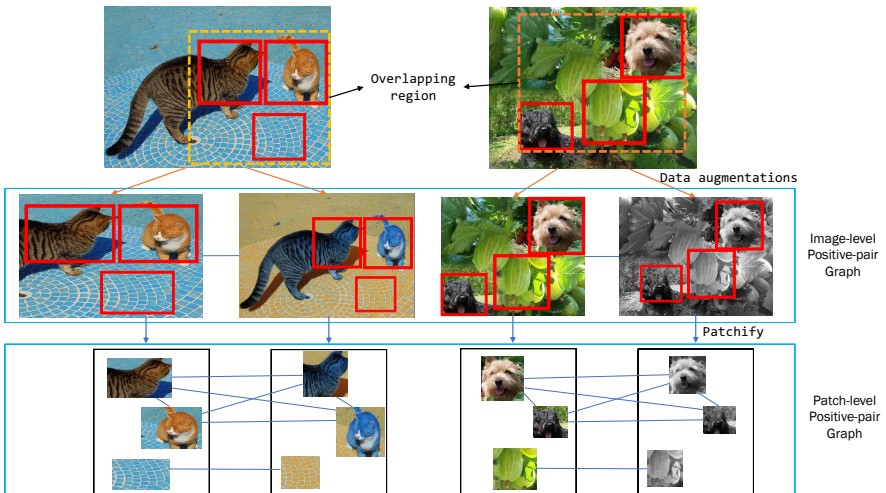

Figure 1: The construction of the hierarchical patch-level positive-pair graph. First, the images are mapped into augmented views by random augmentations, and they form an image-level positive-pair graph (the middle layer). Then, we draw several patches with exact positional correspondence (red boxes) from the overlapping region (orange boxes). By determining the strategy for selecting positive pairs in a single view (i.e., the edges inside the black boxes of the bottom layer), we can generate a patch-level positive-pair graph on top of the image-level positive-pair graph (the bottom layer).

the consistency of $A_I$, the factors that remain are $\alpha$ and the eigenvalues of $\bar{B}$. Intuitively, when more positive pairs are matched, the probability of a positive pair belonging to different classes increases, leading to a higher error rate $\alpha$. Hence, studying the eigenvalue distribution of the normalized strategy matrix $\bar{B}$ can provide insights into how positive pairs impact the model's performance. In the following sections, we begin by providing a theoretical discussion of $B$ as a circulant matrix, corresponding to the scenario of matching spatially close patches into positive pairs under circular padding. Subsequently, we empirically investigate the case where $B$ is a general matrix.

**Eigenvalue Distribution of Circulant Matrix.** First, we discuss the eigenvalue distribution when the strategy matrix $B$ is in the form of a circulant matrix as below.

**Assumption 2.7.** *We consider $B$ to have the following form of a circulant matrix*

$$B_l = \begin{pmatrix} 1 & b_1 & \cdots & b_{m-2} & b_{m-1} \\ b_{m-1} & 1 & b_1 & & b_{m-2} \\ \vdots & b_{m-1} & 1 & \ddots & \vdots \\ b_2 & & \ddots & \ddots & b_1 \\ b_1 & b_2 & \cdots & b_{m-1} & 1 \end{pmatrix}, \tag{8}$$

*where we let $b_1 = \cdots = b_l = b_{m-l} = \cdots = b_{m-1} = 1$, and $b_{l+1} = \cdots = b_{m-l-1} = 0$.*

To illustrate this assumption further, we provide an example in Figure 2. The "1" elements in the major diagonals represent patch pairs that are spatially close, while those in the lower-left and upper-right represent patch pairs that are close due to circular padding. In this case, the normalized strategy matrix $\bar{B}$ exhibits favorable properties, leading to the following theorem:

**Theorem 2.8.** *Assume that $B_l$ satisfies Assumption 2.7 and $\lambda_{i'} > 0$ where $\gamma_{k+1} = \lambda_{i'}\mu_{j'}$ according to the decomposition in Lemma 2.6. Then when $2lj' < m$, we have*

$$\mathcal{E}(f) \lesssim \frac{\alpha}{h(l)} \tag{9}$$

*where $h(l)$ monotonically increases w.r.t. l.*

Since $l$ represents the number of positive pairs, matching more positive pairs will increase the denominator in equation 23 and force the error bound to be smaller. However, as the number of positive pairs increases, the spatial distance of a pair will simultaneously increase. Intuitively, as the spatial distance between patches grows, the probability of them being semantically close decreases.

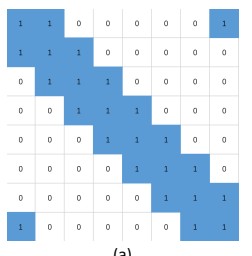 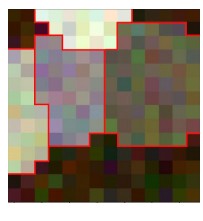 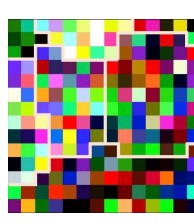

(a)  (b)

Figure 2: **(a)** An example matrix for the Assumption 2.7. Here we consider the case with $m = 8$ patches in the image, and we regard the nearest neighbors with distance $l = 1$ as positive pairs for DCL under circular padding. **(b)** Demonstration of the synthetic images. The two images are generated by the same label (which is annotated in the images) with different noise scale $\tilde{\sigma}$ (**Left**: $\tilde{\sigma} = 0.05$; **Right**: $\tilde{\sigma} = 0.5$). The images are shown by linearly scaling the pixel values into $[0, 1]$.

This phenomenon results in a higher labeling error rate $\alpha$. Therefore, there is a **trade-off** when considering the error bound in relation to the number of positive pairs. On one hand, increasing the number of positive pairs will decrease the eigenvalues of the normalized strategy matrix $\bar{B}$, leading to a smaller error bound. On the other hand, as the number of positive pairs increases, the error rate $\alpha$ also increases, which results in a larger error bound. Thus, finding the right balance between these two factors is crucial for achieving optimal performance in selecting positive pairs.

**Eigenvalue Distribution of General Matrix.** To illustrate the eigenvalue distribution of a general adjacency matrix, we conduct simulation experiments to find that the eigenvalue $\mu_j$ is negatively correlated with the number of positive pairs (details can be found in the Appendix). Hence, in the general condition, the trade-off still exists.

## 2.5 ANALYZING THE TRADE-OFF OF POSITIVE PAIR SELECTION ON SYNTHETIC DATA

To further illustrate the trade-off between the number of positive pairs and the error rate $\alpha$, we conduct a series of experiments on the synthetic dataset.

**Dataset Settings.** The synthetic dataset consists of 1024 training and 256 validation images. Each image has 3 channels and 14x14 pixels, with each pixel belonging to one of the $r + 1$ classes, one of which is the background class (class 0) and the other r classes are the entity classes (class 1 to $r$).

To generate an image, we first generate its label by a Gaussian Mixture Model (GMM) to simulate the distribution of the objects in images. Assume that there are $c$ non-intersecting objects in an image. We define a generating function for each object as

$$g_i(x|\mu_i, \sigma_i) = \frac{1}{2\pi\sigma_i^2} \exp\left(-\frac{\|x - \mu_i\|_2^2}{2\sigma_i^2}\right), \quad i = 1, \cdots, r. \tag{10}$$

Here $x$ is a 2-dimensional coordinate vector on the image which is normally distributed with mean $\mu_i$ and variance $\sigma_i^2 I$. The parameters $\mu_i$ and $\sigma_i$ are respectively i.i.d. random variables. In particular, here we make $\mu_i$ uniformly distributed on the image and $\sigma_i \sim U([a, b])$ to create multiscale objects, where $U$ stands for uniform distribution and $a$ and $b$ are hyperparameters. For the background class, the generating function is defined as $g_0(x) = \beta_0$ where $\beta_0$ is a given threshold.

Then given the random sampled $\{\mu_i\}, \{\sigma_i\}$, the label of a pixel $x$ (denoted as $y(x)$) is determined as

$$y(x) = \underset{i=0,1,\cdots,r}{\arg\max} \, g_i(x|\mu_i, \sigma_i). \tag{11}$$

After obtaining labels for pixels, we can generate the pixel values. Note that in reality, objects belonging to the same class often share similar pixels and features. From this perspective, we utilize another Gaussian model to generate pixel values. For a pixel $x$, the pixel value $p(x) \in \mathbb{R}^3$ is normally distributed as $p(x) \sim \mathcal{N}(p; \tilde{\mu}_{y(x)}, \tilde{\sigma}^2 I)$. Here $\tilde{\mu}_{y(x)} \sim [U([a_{y(x)}, b_{y(x)}])]^3$ represent the center of the object feature where $(a_{y(x)}, b_{y(x)})$ are non-intersecting intervals, and $\tilde{\sigma}$ is a fixed hyperparameter representing the noise magnitude.

Intuitively, for a single pixel, its values are affected by random noise so that sometimes they may fall to intervals of other classes. However, the effect of noise can be mitigated by combining information about pixels in the same class (likely to be close to the pixel itself), for example, by simply taking the

Table 1: DCL on the synthetic dataset with different numbers of positive pairs. $l$ represents that a pixel forms positive pairs with the pixels in its $l \times l$ neighborhood, and $\tilde{\sigma}$ stands for **the noise of pixel values** of the dataset. $l = 27$ means that every pixel pair in an image is considered as a positive pair.

|  | ACC($\tilde{\sigma}$=0.05) | ACC($\tilde{\sigma}$=0.2) | ACC($\tilde{\sigma}$=0.5) | $\alpha$ | $\mu_{k+1}$ | $2\alpha/(1-\mu_{k+1})$ |
|---|---|---|---|---|---|---|
| SUP | **98.4±0.1** | 70.0±1.4 | 52.9±0.1 | - | - | - |
| $l = 1$ | 95.9±0.5 | 71.2±0.4 | 53.7±0.1 | 0 | 1 | UNDEFINED |
| $l = 3$ | 97.4±0.3 | 72.3±0.2 | **54.0±0.1** | 0.14 | 0.59 | 0.68 |
| $l = 5$ | 97.7±0.2 | **72.4±0.1** | 53.8±0.3 | 0.27 | 0.18 | 0.66 |
| $l = 27$ | 77.6±5.1 | 66.4±0.7 | 52.4±1.1 | 0.71 | 0 | 1.42 |

average. Therefore, this setup for generating images is reasonable. In our experiments, we set $r = 4$, $a_i = 0.2 * i$, $b_i = 0.2 * i + 0.2$. For the sake of class balance, each image contains one object of each class. Figure 2 demonstrates the synthetic images along with their labels. We can see that when the noise scale $\tilde{\sigma}$ is large enough, we can hardly discriminate the pixel label from the image.

**Training and Evaluation.** We train a 1-layer ViT on the synthetic dataset with the DCL method to learn unsupervised features of pixels, and evaluate them by training a linear probing for the semantic segmentation task, where the pixel labels are given in the dataset. In the pre-training, for each pixel, we match its $l \times l$ neighborhood as its positive pairs, and other pixels in the same image as negative pairs. Moreover, we calculate the error rate $\alpha$ and the (k+1)-th largest eigenvalue of the normalized strategy matrix $\bar{B}$, $\mu_{k+1}$, where the feature dimension $k$ is 24. Since it is hard to simulate an image-level positive-pair graph by a simple dataset, we calculate $2\alpha/(1-\mu_{k+1})$ to estimate the magnitude of the error bound in equation 4, where the denominator can characterize the monotonic property of the eigenvalue term in the error bound.

**Results.** In Table 1, "Sup" stands for supervised pre-training with the same backbone and evaluation with the same metric. For supervised pre-training, when the noise scale is low, it performs better than unsupervised models. However, when the noise becomes stronger, it gradually becomes difficult to perform the task since the labels can be hardly discriminated by the raw input of single pixels. On the contrary, the DCL methods, especially when $l = 3$ and 5, combine the information of the neighborhood of a pixel, thus learning better representations for pixels.

Among the DCL methods, the error bounds of $l = 3$ and 5 are similar, which are much lower than that of $l = 27$. In particular, the error bound of $l = 1$ is undefined since it is the form of $0/0$, thus out of control. The performance of these methods are consistent with their error bound, with $l = 3$ and 5 being the best in all the noise settings. Although $l = 27$ has the most number of positive pairs, there is too much noise among these pairs, resulting in the bad representations learned. This shows that to get a lower error bound, a method should have more positive pairs but less error rate at the same time.

## 3 COMPATIBILITY OF OUR THEORETICAL ANALYSIS WITH EXISTING METHODS

In this section, we show that our theoretical analysis is compatible with a number of existing works. We take the following DCL methods as examples.

PixPro (Xie et al., 2021) is a classical DCL method. It matches two patches from two augmented views of the same image as positive pairs when the distance of their corresponding positions in the original image is smaller than a given threshold $\mathcal{T}$. Although the positive pairs do not share the exact same position and the matching criterion is not consistent with all views, we argue that its strategy of selecting positive pairs with a threshold $\mathcal{T}$ is aimed at aligning the spatial position of patches. Thus it can still be applied to our framework. Additionally, it introduces a PPM module to map the feature of a pixel into the linear aggregation of all the pixels in the same feature map. Since the loss function is the inner product of two features, aligning the feature of a pixel $p$ with the PPM output of another pixel $p'$ is equivalent to aligning the feature of $p$ and the features of all the pixels in the feature map of $p'$ with different weights by the back-propagation. Although this module dynamically matches positive pairs, we argue that this is not the crucial part of PixPro, since, in the following experiment, we show that without the PPM module, we can still obtain a similar performance with PixPro.

DUPR (Ding et al., 2022) uses the RoI Align module to solve the problem of misaligned patches. In this method, the pixels of all layers of the ConvNet are involved in DCL. Hence, in the perspective of our framework, the set of the augmented patches consists of the pixels of all the layers after applying RoI Align. Moreover, since DUPR only matches patches that have exactly the same position as a

Table 2: We evaluate different DCL models on different downstream tasks. Specifically, we report AP for Pascal VOC (Everingham et al., 2009) object detection task and report $AP_{mk}$ and $AP_{bb}$ for COCO (Lin et al., 2014) object detection and instance segmentation, respectively. All models are pre-trained for 100 epochs on the ImageNet (Deng et al., 2009) and evaluated with Faster R-CNN (R50-C4) (Ren et al., 2015) on PASVAL VOC task and Mask R-CNN (R50-FPN) (He et al., 2017) on COCO tasks with 1× schedule. *: copied from Xie et al. (2021).

| METHOD | LOSS | VOC DET. | COCO DET. | COCO SEG. |
|---|---|---|---|---|
| PIXCONTRAST* (XIE ET AL., 2021) | INFONCE | 58.1 | 38.8 | - |
| PIXPRO (XIE ET AL., 2021) | MSE | 58.8 | 40.8 | 36.8 |
| PIXCONTRAST | SPECTRAL | 59.0 | 42.0 | 38.1 |

positive pair, the strategy matrix $B$ is an identity matrix. Thus our theoretical results can be applied to this work. SoCo (Wei et al., 2021) generates the patches directly on the natural images using selective search. They further downsample the images to obtain multi-scale patches. Similar to DUPR, they also utilize the feature maps of different layers and match positive pairs that share the same position. Thus our theoretical results can also be applied to this method.

Among the ViT-based methods, recent models like iBOT (Zhou et al., 2021), ADCLR (Zhang et al., 2022), and PQCL (Zhang et al., 2023) can all be applied to our work. Specifically, iBOT employs masked image modeling (MIM) on the augmented views and aligns patch features through self-distillation; ADCLR introduces query patches for patch-level contrasting and leverages the cross-attention mechanism to produce different encodings of the same query patch, and matches them as positive pairs; PQCL incorporating cross-attention in each block between positional and patch embeddings to encode the query patches to different features as positive pairs. It is noteworthy that all three methods achieve exact 1-to-1 patch correspondence. The essence of these methods is to generate different encodings of the same patch, which are used to form positive pairs. Same as DUPR, these approaches correspond to $B = I$ in our paper due to the exact patch correspondence.

**Verification on the Compatibility of Our Framework.** To further verify the compatibility of our framework, we test the effect of spectral dense contrastive learning with PixContrast (Xie et al., 2021), a simplified version of PixPro. Compared with PixPro, PixContrast does not use the PPM module to aggregate the features of patches, and it takes InfoNCE (Oord et al., 2018) as its loss, unlike PixPro which uses MSE loss and only utilizes positive pairs. We simply replace the InfoNCE with spectral contrastive loss, and apply the same process of dense contrastive pre-training and downstream transferring for both methods. We follow the same training settings in the paper. The results are shown in Table 4. We surprisingly find that the spectral PixContrast is much better than the standard version, and even better than PixPro, which has an additional well-designed PPM module. This verifies both the effectiveness of the spectral DCL and the compatibility of our framework.

## 4    SURROGATE METRICS FOR POSITIVE PAIR SELECTION

The error bound for DCL (Equation 4) is affected by the error rate of positive pairs $\alpha$ and the eigenvalues of the normalized patch-level adjacency matrix $\bar{A}$. However, both quantities can hardly be measured in the unsupervised pre-training process since $\alpha$ relies on the ground truth and the eigenvalues are intractable. Although we have previously shown a trade-off between these two quantities in section 2.4, we still do not know how to select better positive pairs. As a result, we propose two metrics for each quantity that can be directly computed using pre-trained features and do not require patch labels. These metrics allow us to easily determine which strategy of selecting positive pairs is superior.

**Metric for Eigenvalue.** Due to the large size of the normalized patch-level adjacency matrix $\bar{A}$, calculating its eigenvalues directly is infeasible. However, we can use the number of positive pairs as an approximate estimate of the eigenvalues, or equivalently, the connectivity of the patch-level positive-pair graph. Taking inspiration from the Average Relative Confusion (ARC) metric in (Wang et al., 2022b), we define a new locality-aware surrogate metric named Patch Confusion Rate (PCR). Its definition is given as follows. Let $p$ be an arbitrary patch in a view $x$. Let $p' = NN(p)$ be the nearest neighbor of $p$ in the set of all patches $\mathcal{X}_p$ in the feature space. Then the PCR is calculated as $PCR(\mathcal{X}_P) = \mathbb{E}_{p \in \mathcal{X}_P} \mathbf{1}(p \subset x, p' \subset x)$. It represents the ratio of nearest neighboring patches that originate from the same image as the anchor patch, considering all patch features in the dataset. Thus, a lower PCR indicates stronger connectivity between patches in the dataset, which in turn implies better generalization according to our theory.

Table 3: The performance of different positive pair selecting strategies of PixPro/PixContrast and their PCR and loss over the pre-trained features. The term "Strategy" denotes the strategy of selecting positive pairs, which is the scope of neighbors within which a patch is matched with its positive pairs. (The term "PPM" refers to the original PixPro). "PixCon." stands for PixContrast. *: copied from Xie et al. (2021).

| MODELS | STRATEGY | VOC DET. | COCO DET. | COCO SEG. | PCR($\downarrow$) | LOSS($\downarrow$) |
|---|---|---|---|---|---|---|
| PIXPRO | PPM | 58.8 | 40.8 | 36.8 | 0.80 | -3.58 |
| SIMPLE PIXPRO | $1\times1$ | 56.5 | 40.7 | 36.5 | 0.95 | -3.60 |
| | $3\times3$ | 56.9 | 39.2 | 35.4 | 0.59 | -2.08 |
| | $5\times5$ | 53.9 | 37.3 | 33.9 | 0.15 | -0.92 |
| | ALL | 54.1 | 39.3 | 35.7 | 0.83 | -1.44 |
| PIXCON. * | $1\times1$ | 58.1 | 38.8 | - | - | - |
| SPECTRAL PIXCON. | $1\times1$ | 59.0 | 42.0 | 38.1 | 0.93 | -2.06 |
| | $3\times3$ | 57.2 | 38.2 | 34.8 | 0.28 | -0.78 |
| | $5\times5$ | 55.8 | 37.8 | 34.1 | 0.11 | -0.55 |
| | ALL | 55.1 | 38.0 | 34.5 | 0.84 | -0.83 |

**Metric for Error Rate.** The error rate of positive pairs, $\alpha$, is a quantity decided by both the positive pairs and the ground truth. However, the ground truth is not accessible in pre-training. We directly consider the patch-level contrastive loss calculated on the pre-trained features as the metric for the error rate. For most models, the contrastive loss measures the feature similarity between positive pairs, and we regard the feature similarity as the probability of a positive pair belonging to the same class. Therefore, the contrastive loss of the model describes the scale of the error rate $\alpha$ to a certain extent, i.e., the smaller the loss function over pre-trained features, the smaller the error rate $\alpha$.

**Experiments.** We conducted experiments to verify the effectiveness of the suggested metrics. To compare the performance of the two metrics under different strategies for selecting positive pairs, we modified the Pixel Propagation Module (PPM) in our base method PixPro. Instead of using a weighted average process, we used a simple average of the features from neighboring patches of size $N \times N$. This is equivalent to matching a patch with its $N \times N$ neighbor patches, as mentioned in Section 3. By adjusting the value of $N$, we could control the number of positive pairs selected. We also performed similar experiments on PixContrast with spectral loss using the same approach.

The results are shown in Table 3. For all these settings of positive pairs, the two metrics give a good indication of the performance of the models. For example, among simple PixPro methods, compared with the setting of "All", the "$3\times3$" setting has both a smaller PCR and lower loss, so its error bound is smaller, resulting in +2.8AP on the PASCAL VOC dataset; compared with the "$1\times1$" setting, the standard PixPro model has a similar loss but a smaller PCR, which leads better performance on all the downstream tasks, especially with a +2.2AP performance on the PASCAL VOC dataset. In addition, this experiment also shows the trade-off between the two quantities: compared with the "$5\times5$" setting of Spectral PixContrast, the "All" setting has a larger PCR but smaller loss, resulting in the two models performing similarly. Additionally, to show the effectiveness of the proposed metrics on other DCL methods, we conduct experiments on SoCo (Wei et al., 2021), which is in Appendix B.

With these two metrics, we can compare different positive pair selection strategies based on the learned representations. By evaluating their effectiveness, we can determine which strategy performs better according to the metrics. The metrics can also be used as a filtering metric to select positive pairs that are more label-consistent, indicated by a smaller value of $\alpha$. This can help improve the label consistency and overall performance of the model. In the experiments on SoCo (in the appendix), we also demonstrate how the metrics can be used to compare models with different training settings such as training epochs. However, the specific design of the model itself is left for future work.

## 5    CONCLUSION

In this paper, we introduce a theoretical framework for dense contrastive learning utilizing the patch-level positive-pair graph. Building upon this graph, we conduct theoretical analyses that encompass performance error bounds and the impact of positive pairs. Moreover, by decoupling the image-level and patch-level supervision in dense contrastive learning, we discover a performance trade-off between the quantity and correctness of positive pairs. Lastly, we propose two unsupervised metrics to guide the selection of positive pairs in dense contrastive learning.

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

## A SIMULATION EXPERIMENT OF THE EIGENVALUE DISTRIBUTION OF A GENERAL MATRIX

For the intra-view adjacency matrix $B$, we initialize it as an identity matrix. In each iteration, we randomly change a 0 element to 1. Figure 3 illustrates the evolution of eigenvalues of $\bar{B}$ during this process. The eigenvalues are obtained by repeating the experiment 20 times. In the left figure, we set the dimension of $B$ to be 25, while in the right figure, we set it to be 50. As observed, the eigenvalues generally decrease as the number of 1 elements in $B$ increases.

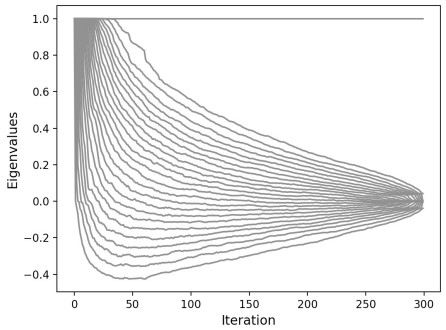 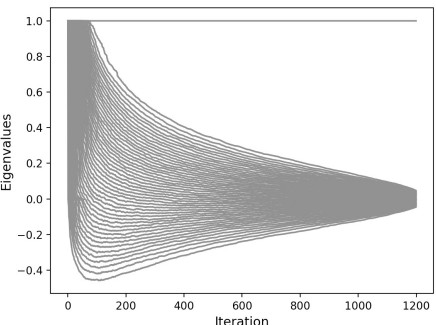

Figure 3: Eigenvalues of the simulated normalized strategy matrix $\bar{B}$.

## B ADDITIONAL ABLATION STUDIES ON THE EFFECTIVENESS OF THE PROPOSED METRICS

In this section, we evaluate the effectiveness of the proposed metrics on another DCL method, i.e., SoCo (Wei et al., 2021). As discussed in section 3, this method is also compatible with our theory.

Specifically, we evaluate the proposed metrics on two released pre-trained models of SoCo using different training epochs. Note that changing pre-training epochs will not affect the augmentation class and the training loss, thus the proposed metrics are still applicable in principle. The results are shown below. Even without knowing the information of the training epoch, we can still tell that the pre-trained model in the bottom performs better from the metrics, since it has smaller PCR and lower loss.

Table 4: The evaluation performance along with the two proposed metrics (i.e., PCR and loss) of two released pre-trained SoCo models. The two models differ in their training epochs (100 vs. 400), and we test its COCO object detection and instance segmentation performance using Mask-RCNN backbone, which is copied from its original paper.

| METHOD | EPOCHS | $AP_{bb}$ | $AP_{mk}$ | PCR | LOSS |
|--------|--------|-----------|-----------|-----|------|
| SoCo | 100 | 42.3 | 37.6 | 0.83 | -7.80 |
|  | 400 | 43.0 | 38.2 | 0.77 | -7.90 |

## C PROOFS

### C.1 PROOF OF THEOREM 2.3

We first give the concrete definition of the error term as below. Let a linear classifier has weight $C \in \mathbb{R}^{k \times r}$ and predicts $g_{f_I,C}(\bar{x}) := \arg\max_{i \in [r]} (f_I(x)^\top C)_i$ for an augmented view $x$. Given a natural image sample $\bar{x}$, we ensemble the predictions on augmented views and predict $\bar{g}_{f_I,C}(\bar{x}) := \arg\max_{i \in [r]} \Pr_{x \sim \mathcal{A}(\cdot|\bar{x})} [g_{f_I,C}(x) = i]$. Then the linear probe error, i.e., the error of the best possible linear classifier on the representations is defined as $\mathcal{E}(f_I) := \min_{C \in \mathbb{R}^{k \times r}} \Pr_{\bar{x} \sim \mathcal{P}_{\overline{\mathcal{X}}}} [y(\bar{x}) \neq \bar{g}_{f_I,C}(\bar{x})]$. In the following, we restate Theorem 2.3.

**Theorem C.1.** *We can upper-bound the linear probe error $\mathcal{E}(f_I)$ as*

$$\mathcal{E}(f_I) \lesssim \frac{4\varepsilon}{1 - \lambda_{k+1}}. \tag{12}$$

*where $\lambda_{k+1}$ is the $(k+1)$-th largest eigenvalue of $\bar{A}_I$.*

*Proof.* In HaoChen et al. (2021), the error bound is given as

$$\mathcal{E}(F) \lesssim \frac{2\phi^{\hat{y}}}{1 - \lambda_{k+1}}, \tag{13}$$

where $\phi^{\hat{y}_I} = \sum_{x,x'\in\mathcal{X}} w_{xx'} \cdot \mathbb{1}\left[\hat{y}_I(x) \neq \hat{y}_I(x')\right]$, and $\hat{y}_I$ denotes the extended label of augmented views. This quantity characterizes the difference in semantics between two augmented views of the same natural image. Combining the fact that $\phi^{\hat{y}_I} \leq 2\varepsilon$ (Lemma C.5 in HaoChen et al. (2021)), we can derive equation 12 in Theorem 2.3. □

## C.2 PROOF OF THEOREM 2.4

To give the formal expression of the linear probe error in the context of dense contrastive learning, we first introduce some notations. We denote $\bar{p}$ as a patch in the natural image. In our modeling of the process of selecting positive pairs, every $\bar{p}$ has a natural correspondence to the patches in the augmentation views.

We denote "$p$ is a random patch from the view $x$" as $x \to p$. When $\bar{x} \to \bar{p}$ and $x \sim \mathcal{A}(\cdot|\bar{x})$, we denote "$p$ is the corresponding patch of $\bar{p}$" as $p \sim \bar{p}$. Let a linear classifier be applied on a patch feature with weights $C \in \mathbb{R}^{k \times r}$. With this, a patch $p$ is predicted as $g_{f,C}(p) = \arg\max_{i\in[r]}(f(p)^\top C)_i$. Then, for a patch $\bar{p}$ from a natural data sample $\bar{x}$, the predictions of all its augmentations are aggregated as

$$\bar{g}_{f,C}(\bar{p}) = \arg\max_{i\in[r]} \Pr_{\substack{x\sim\mathcal{A}(\cdot|\bar{x})\\ \bar{x}\to\bar{p},x\to p,p\sim\bar{p}}} [g_{f,C}(p) = i]. \tag{14}$$

With these, we define the linear probe error of patches as

$$\mathcal{E}(f) := \min_{C\in\mathbb{R}^{k\times r}} \Pr_{\substack{\bar{x}\sim\mathcal{P}_{\overline{\mathcal{X}}}\\ \bar{x}\to\bar{p}}} [y(\bar{p}) \neq \bar{g}_{f,C}(\bar{p})]. \tag{15}$$

In the following, we restate Theorem 2.4.

**Theorem C.2.** *Define the labeling error rate $\alpha$ as the average label disagreement among the positive pairs $(p, p^+)$, i.e., $\alpha := \mathbb{E}_{p,p^+}\mathbb{1}[y(p) \neq y(p^+)]$. We can upper-bound the linear probe error for DCL as*

$$\mathcal{E}(f) \lesssim \frac{2\alpha}{1 - \gamma_{k+1}}. \tag{16}$$

*where $\gamma_{k+1}$ is the $(k+1)$-th largest eigenvalue of $\bar{A}$.*

*Proof.* In the case of DCL, the labeling error rate $\alpha$ is defined as the difference between the labels of two patches, which is equivalent to the quantity $\phi^y = \sum_{p,p'\in\mathcal{X}_P} w_{pp'} \cdot \mathbb{1}[y(p) \neq y(p')]$ ($\phi^y$ no longer has connection with $\varepsilon$). As a result, equation 13, which characterizes the error bound in GCL, can be directly applied to DCL. This corresponds to equation 16. □

## C.3 PROOF OF LEMMA 2.5

We restate Lemma 2.5 as below.

**Lemma C.3.** *Let $\bar{(\cdot)}$ be the normalized form of a matrix defined in Section 2.1. Given $A = A' \otimes B$, we have*

$$\bar{A} = \bar{A}' \otimes \bar{B}. \tag{17}$$

(For convenience in notation, here we let $A' = A_I$.)

*Proof.* Recall that the dimension of $A'$ and $B$ are respectively $n$ and $m$. We further denote $A_{i,k;j,l} := A_{im+k,jm+l} = A'_{i,j}B_{k,l}$. By the definition of normalization, we have

$$\bar{A}'_{i,j} = \frac{A'_{i,j}}{\sqrt{\left(\sum\limits_{k=1}^{n} A'_{i,k}\right)\left(\sum\limits_{k=1}^{n} A'_{j,k}\right)}}. \tag{18}$$

The degree of $A$ is

$$\begin{aligned}
\sum_{j,l} A_{i,k;j,l} &= \sum_{l=1}^{m}\sum_{j=1}^{n} A'_{i,j}B_{k,l}\\
&= \sum_{j=1}^{n} A'_{i,j}\sum_{l=1}^{m} B_{k,l}.
\end{aligned} \tag{19}$$

Then we can calculate the element of $\bar{A}$:

$$\begin{aligned}
\bar{A}_{i,k;j,l} &= \frac{A_{i,k;j,l}}{\sqrt{\left(\sum_{j,l} A_{i,k;j,l}\right)\left(\sum_{i,k} A_{i,k;j,l}\right)}}\\
&= \frac{A'_{i,j}B_{k,l}}{\sqrt{\left(\sum\limits_{j=1}^{n} A'_{i,j}\sum\limits_{l=1}^{m} B_{k,l}\right)\left(\sum\limits_{i=1}^{n} A'_{i,j}\sum\limits_{j=1}^{m} B_{k,l}\right)}}\\
&= \bar{A}'_{i,j}\bar{B}_{k,l}.
\end{aligned} \tag{20}$$

This exactly the definition of $\bar{A} = \bar{A}' \otimes \bar{B}$. □

## C.4 PROOF OF LEMMA 2.6

We restate Lemma 2.6 as below.

**Lemma C.4.** Let $A_1 \in \mathbb{R}^{n \times n}$ have eigenvalues $\lambda_i, i \in [n]$, and let $A_2 \in \mathbb{R}^{m \times m}$ have eigenvalues $\mu_j, j \in [m]$. Then the $mn$ eigenvalues of $A_1 \otimes A_2$ are $\lambda_1\mu_1, \ldots, \lambda_1\mu_m, \lambda_2\mu_1, \ldots, \lambda_2\mu_m, \ldots, \lambda_n\mu_m$.

*Proof.* The Kronecker product has the following property:

$$(A_1 \otimes A_2)(x \otimes z) = A_1 x \otimes A_2 z, \tag{21}$$

where $A_1 \in \mathbb{R}^{n \times n}, A_2 \in \mathbb{R}^{m \times m}, x \in \mathbb{R}^n, z \in \mathbb{R}^m$. Then if $x$ is the right eigenvector of $A_1$ with eigenvalue $\lambda$, and $z$ is the right eigenvector of $A_2$ with eigenvalue $\mu$, we have

$$\begin{aligned}
(A_1 \otimes A_2)(x \otimes z) &= A_1 x \otimes A_2 z\\
&= \lambda x \otimes \mu z\\
&= \lambda\mu(x \otimes z).
\end{aligned} \tag{22}$$

Thus, $\lambda\mu$ is an eigenvalue of $A \otimes B$, which finishes the proof. □

## C.5 PROOF OF THEOREM 2.8

We restate Theorem 2.8 as below.

**Theorem C.5.** Assume that $B_l$ satisfies Assumption 2.7 and $\lambda_{i'} > 0$ where $\gamma_{k+1} = \lambda_{i'}\mu_{j'}$ according to the decomposition in Lemma 2.6. Then when $2lj' < m$, we have

$$\mathcal{E}(F) \lesssim \frac{\alpha}{h(l)} \tag{23}$$

where $h(l) > 0$ monotonically increases w.r.t. l.

*Proof.* The eigenvalues of a general circulant matrix (in the form of equation 8) is well-known as

$$\lambda_k = f(\omega_k), \; k = 0, 1, \cdots, m-1, \tag{24}$$

where $f(x) = b_0 + b_1 x + b_2 x^2 + \cdots + b_{m-1} x^{m-1}$ with $b_0 = 1$, and $\omega_k = \cos \frac{2k\pi}{m} + i \sin \frac{2k\pi}{m}$. Then when $b_{l+1} = \cdots = b_{m-l-1} = 0$ and $b_1 = \cdots = b_l = b_{m-l} = \cdots = b_{m-1} = 1$, the eigenvalues of $\bar{B}_l$ are

$$\mu_{l,k} = \frac{1 + 2 \sum_{j=1}^{l} \cos \frac{2jk\pi}{m}}{1 + 2l}, \ k = 0, 1, \cdots, m-1. \tag{25}$$

It is evident that when $2kl < m$, the eigenvalue $\mu_{l,k}$ monotonically decreases w.r.t. $l$. Furthermore, assuming the consistency of $\{\lambda_i\}$, it can be observed that as every eigenvalue in $\{\mu_j\}$ decreases, the eigenvalue $\gamma_{k+1}$ also decreases. Consequently, $\gamma_{k+1}$ monotonically decreases with respect to $l$, thereby completing the proof. □

## D  SUPPLEMENTARY INTRODUCTION OF SPECTRAL CONTRASTIVE LEARNING

In this part, we first describe the process of spectral clustering and then prove the equivalence between spectral clustering and spectral contrastive learning.

In the mathematical formulation described in Section 2.1, let $v_1, v_2, \cdots, v_k$ be the corresponding unit-norm eigenvectors of the $k$ largest eigenvalues $\gamma_1, \gamma_2, \cdots, \gamma_k$. By stacking the eigenvectors in columns, we obtain the matrix $F^\star = [v_1, v_2, \cdots, v_k] \in \mathbb{R}^{n \times k}$. The rows of $F^\star$ can be treated as embeddings of each vertex in the graph $G_I$, which possess good clustering properties. Typically, spectral clustering algorithms apply K-Means to these embeddings to obtain the clusters.

However, due to the exponentially large size of the patch-level positive-pair graph, it is infeasible to store the embedding $F^\star$ directly. Instead, we parameterize the embeddings as a neural network function $f : \mathcal{X} \to \mathbb{R}^k$ and optimize it using the low-rank approximation:

$$\min_{F \in \mathbb{R}^{n \times k}} \mathcal{L}_{\mathrm{mf}}(F) = \|\bar{A} - FF^\top\|_F^2, \tag{26}$$

where the rows of $F$ correspond to the embeddings calculated by the network. In this scenario, optimizing the network using the matrix factorization loss $\mathcal{L}_{\mathrm{mf}}(F)$ is equivalent to using the spectral contrastive loss (equation 1), i.e., $\mathcal{L}_{\mathrm{mf}}(F) = \mathcal{L}(f) + \mathrm{const}$. Detailed proofs of this equivalence can be found in HaoChen et al. (2021).

## E  RELATED WORK

**Dense Contrastive Learning.** The DCL methods can be divided into two main streams regarding their strategies for matching positive patch pairs: feature-based and geometry-based methods (Wang et al., 2022c). Feature-based methods align patches based on feature similarity. For instance, DenseCL (Wang et al., 2021) establishes patch correspondence by computing cosine similarities between the feature maps of patches. These methods typically match positive pairs dynamically, which can result in the challenge of unstable pre-training during early epochs. Geometry-based methods match patch pairs based on their distance in the original images, which have been more frequently considered in recent studies and are our main focus in this paper. For instance, PixPro (Xie et al., 2021) determines positive pairs by the distance between patches, using a threshold. DUPR (Ding et al., 2022) aligns two misaligned views by applying RoI Align on their intersection, which naturally produces positive pairs without hyperparameters. SoCo (Wei et al., 2021) aligns patches in various regions and scales to achieve object-level translation and scale invariance. Recently, several ViT-based DCL methods have emerged. By generating different encodings of the same patch to form positive pairs, they all achieve precise correspondence in geometry, leading to outstanding performance (Zhou et al., 2021; Zhang et al., 2022; 2023).

