# OpenReview forum: "On the Theoretical Analysis of Dense Contrastive Learning"
_ICLR.cc/2024/Conference — Submitted to ICLR 2024_

### Official Review · Reviewer_hTKu · 2023-10-23

**Soundness:** 2 fair
**Presentation:** 2 fair
**Contribution:** 2 fair
**Rating:** 6
**Confidence:** 3

**Summary:**

This paper uses the language of spectral graph theory to establish the first theoretical framework for modeling and analyzing DCL by dissecting the corresponding patch-level positive-pair graph. Specifically, by decoupling the image-level and patch-level supervision, the authors theoretically characterize how different positive pair selection strategies affect the performance of DCL, and verify these insights on
both synthetic and real-world datasets.

**Strengths:**

The paper is well-organized and easy to follow.

Using the Kronecker product to decompose the adjacency matrix of the patch-level positive-pair graph is novel and make sense.

**Weaknesses:**

1. Missing important references about DCL methods (iBOT, ADCLR, PQCL).

[1] Zhou J, Wei C, Wang H, et al. ibot: Image bert pre-training with online tokenizer[J]. ICLR 2022.

[2] Zhang S, Zhu F, Zhao R, et al. Patch-level contrasting without patch correspondence for accurate and dense contrastive representation learning[J]. ICLR 2023.

[3] Zhang S, Zhou Q, Wang Z, et al. Patch-level Contrastive Learning via Positional Query for Visual Pre-training[J]. ICML 2023.

**Questions:**

I have no further questions.

---

> ### Author Response · Authors · 2023-11-19
>
> # Response to Reviewer hTKu
>
> We thank Reviewer hTKu for providing constructive comments and appreciating the solidness of our work. Now we address your concerns.
>
> ---
>
> **Q1.** Missing important references about DCL methods (iBOT, ADCLR, PQCL).
>
> **A1.** Thanks for pointing out these relevant pieces of literature. Indeed all three methods are DCL approaches built upon the ViT architecture. Specifically, iBOT employs masked image modeling (MIM) on the augmented views and aligns patch features through self-distillation; ADCLR introduces query patches for patch-level contrasting and leverages the cross-attention mechanism to produce different encodings of the same query patch, and matches them as positive pairs; PQCL incorporating cross-attention in each block between positional and patch embeddings to encode the query patches to different features as positive pairs.
>
> In fact, all these mentioned works can be aligned with our theoretical framework. It's noteworthy that all three methods achieve exact 1-to-1 patch correspondence. Similar to DUPR, these approaches correspond to $B=I$ in our paper due to the exact patch correspondence. We believe that our proposed insights, such as the suggestion to increase image-level connectivity, can still be applicable to these methods. Besides, while these three works propose effective algorithms, they do not provide theoretical analysis as in our work.
>
> Relative discussion is added in the PDF (Page 8 and Appendix E).

---

> > ### Comment · Reviewer_hTKu · 2023-11-20
> > **Thanks for the authors' response**
> >
> > After reading the authors' response, I keep my rating.

---

### Official Review · Reviewer_MfsU · 2023-10-31

**Soundness:** 2 fair
**Presentation:** 1 poor
**Contribution:** 2 fair
**Rating:** 6
**Confidence:** 4

**Summary:**

This paper introduces a theoretical framework for dense contrastive learning (DCL) in computer vision for downstream tasks. While contrastive learning has excelled in image-level matching, DCL explores patch-level contrast for local information. The paper dissects the patch-level positive-pair graph using spectral graph theory and highlights the impact of different positive pair selection strategies on DCL performance. The paper proposes two unsupervised metrics based on their theoretical insights to guide positive pair selection. They empirically validate their framework on synthetic datasets, providing a foundational understanding of DCL's working principles and generalization ability. This work aims at developing principled DCL methods for dense prediction tasks.

**Strengths:**

- The introduction of two unsupervised metrics for guiding the selection of positive pairs is a novel contribution. These metrics offer a practical way to evaluate and fine-tune dense contrastive learning systems.
- The study aligns with current trends in the field of machine learning, particularly in self-supervised and contrastive learning, which adds relevance and interest to the research.

**Weaknesses:**

The main problem of the method seems to be that by switching from natural images to patches, the results from HaoChen et al. (2021) cannot be carried over into DCL.

I make a few points, mainly on the positive pairs selection process and the role of the augmentations distribution {\mathcal A}:

In HaoChen et al. (2021), it is assumed that all natural images \overline{X} have the same distribution \mathcal P_{\overline X},  e.g. a mixture of manifolds, and each image is specified by a single class r. Furthermore, there is a distinction between the population distribution over \overline{X} and the distribution \mathcal A(.|\overline{x}), which is crucial to compute the weights used in the adjacency matrix. The population graph is the graph of all augmentations, including crops. \
Therefore, in HaoChen et al. (2021), wxx’ = E_{\overline{x} \in P_\overline{X}}(A( x| \overline{x}), A( x’| \overline{x})), is a marginal probability of generating a pair (x,x’)  from an image \overline{x}, sampled from P_{\overline{X}).\
If the augmentation would not include cropping, this probability is always 1, because any other augmentations, like rotation, jittering, blurring, etc., do not change the class.

The process of selecting a positive pair is envisioned in this paper in two steps:

1. Select patches on the original image.
2. Do the augmentation, like jittering and transformations.


In equation (3) and (5) it is not clear:

1) if cropping is allowed on the views.
2) If a view includes the choice of a patch.

- If cropping is allowed on views and patches are chosen on the cropped views, then the purpose of B in (5)  is useless since it is not possible to know if the extracted patches are from the same class. As in Figure 1, right, one patch could be a leaf and another a dog head.
- If cropping is not allowed on views and patches are chosen on views, then again, positive pairs are hard in case of random rotations or random stretching since even if on the same location in the different views, they can be of different classes. Then again, B has no purpose, as patch classes cannot be determined.
- if cropping is not allowed and patches are chosen on the natural image, then \mathcal A, namely the probability, given a natural image \overline{x}, that the augmentation x and x’ form a positive pair, makes the problem trivial. In fact, on the image \overline{x}, whatever patch is chosen, the augmentation can stretch them, change colour, blur or rotate them; they will always be of the same class if they come from the same region. Therefore, B is trivial: only patches from different locations can differ.

In this last case, which seems to be the choice of the paper, however, the probability of being a positive pair is still attributed to the views, while it is simply B, the matrix sanctioning the connections. Therefore, it seems that wxx’ is either 1 or 0.

In the proof of Theorem 2.4, it seems that it is possible to choose a patch both from the natural image and a patch from some of its views. \
Then, suppose that \overline{p} is a patch from \overline{x}, and that x is a random crop of \overline{x} that does not include \overline{p}, and p is a patch from x. Then p and \overline{p} could even be of different classes, as argued above.\
Therefore, what is the meaning of the sentence  “p is the corresponding patch of \overline{p}”?   Furthermore, a clear definition of the encoding f for patches is not given, as it is provided only for views (see page 12 Proof of Theorem 2.3); however, in the proof of Theorem 2.4, f(p) is used.

Other observations:

- F is used in many equations (e.g. in equation 1, it should be f) without being suitably introduced.
- In Theorem 2.3, the function f does not appear, though it appears in C Proof, page 12, and it is not said that f should be an embedding function,  and here, too, F remains purposeless. Also, it seems that \psi^y cannot be defined as 2\epsilon since the similarity between augmentations (including cropping) is different from the similarity between views as stated here, where patch/cropping and other augmentations are decoupled.
- The relation between B  as a circulant matrix and the general matrix must be discussed. I could not find an explanation in the appendix.

- In theorem 2.8, it seems that h(l) is not used. Further, the sentence “the spatial distance of a pair will simultaneously increase, which means a larger labeling error rate α” should be made precise.

- D in the appendix comes from HaoChen et al. (2021).
- In Table 2, it probably needs to be noticed that for COCO instance segmentation, Mask R-CNN is used.

No sensible new theory is added. Furthermore,  many statements remain unjustified. The overall writing quality needs improvement in terms of clarity and organization. The authors state that their theoretical framework can be extended to other approaches, but they only experiment on one, namely PixContrast.

**Questions:**

- In the paper, eigenvalue distributions are discussed about graph matrices. How were these distributions simulated, and what significance do they hold for the study?
- Why is the eigenvalue µ_(k+1) of B a good surrogate to compute error? (See Training and Evaluation section).
- What is F in Equation (1)?
- In the explanation on page 7, last paragraph, what is meant in the sentence  “positions in the original image are smaller than a given threshold”?
- Most notably, why strictly follow HaoChen et al. (2021), while the authors have written that it is hard to extend their approach to object detection and segmentation.

---

> ### Author Response · Authors · 2023-11-19
>
> # Response to Reviewer MfsU (1/4)
>
> We thank Reviewer MfsU for providing insightful questions and appreciating the novelty and the theory of our work. We will address your concerns in the following points.
>
> ---
>
> **Q1.** The potential value of $w_{xx'}$ is constrained to either 0 or 1, which appears unreasonable.
>
> **A1.** It is essential to clarify that $w_{xx'} = E_{\overline{x} \sim P_{\overline{\mathcal{X}}}}(A(x|\overline{x}) A(x'|\overline{x}))$ represents the marginal probability of generating a pair $(x,x')$ from an image $\overline{x}$ sampled from $P_{\overline{\mathcal{X}}}$, as noted by the reviewer. Notably, this probability is calculated on data sample $\overline{x}$, but not on the data classes $y \in [r]$. Therefore, this probability remains unaffected by the classes of augmented views or the category of data augmentations. For each $\overline{x} \in P_{\overline{\mathcal{X}}}$, as long as the number of its augmented views exceeds one (a realistic scenario), $0 < A(x|\overline{x}) < 1$ for any of its augmented views $x$. Consequently, $w_{xx'}$ is consistently less than 1, irrespective of the use of cropping.
>
> ---
>
> **Q2.**  In equation (3) and (5) it is not clear: 1) if cropping is allowed on the views. 2) If a view includes the choice of a patch.
>
> - If cropping is allowed on views and patches are chosen on the cropped views, then the purpose of $B$ in (5) is useless since it is not possible to know if the extracted patches are from the same class. As in Figure 1, right, one patch could be a leaf and another a dog head.
> - If cropping is not allowed on views and patches are chosen on views, then again, positive pairs are hard in case of random rotations or random stretching since even if on the same location in the different views, they can be of different classes. Then again, $B$ has no purpose, as patch classes cannot be determined.
> - if cropping is not allowed and patches are chosen on the natural image, then $\mathcal{A}$, namely the probability, given a natural image $\overline{x}$, that the augmentation $x$ and $x’$ form a positive pair, makes the problem trivial. In fact, on the image $\overline{x}$, whatever patch is chosen, the augmentation can stretch them, change colour, blur or rotate them; they will always be of the same class if they come from the same region. Therefore, $B$ is trivial: only patches from different locations can differ.
>
> In this last case, which seems to be the choice of the paper, however, the probability of being a positive pair is still attributed to the views, while it is simply $B$, the matrix sanctioning the connections.
>
> **A2.** We are afraid that the process of selecting a positive pair of our approach does not belong to any of the three reviewer’s assumptions. Instead, **our modeling allows cropping on views, and a view does not include the choice of a patch** (i.e., patches are chosen from **natural** images). Figure 1 illustrates the steps involved in selecting a patch-level positive pair in our model:
>
> 1. Selection of patches from a natural image.
> 2. Application of data augmentations, including cropping, on the natural image.
> 3. During augmentation, the selected patches are mapped to corresponding patches on each augmented view.
> 4. Utilization of a positive pair selection strategy to match these patches into positive pairs.
>
> Furthermore, it is crucial to emphasize that $B$ is defined by the positive pair selecting strategy and holds no connection to the ground-truth class of the patches. As discussed in Section 3, there are methods that establish 1-to-1 correspondence of positive patch pairs, aligning with $B=I$. On the other hand, there are methods that match a patch with multiple patches as positive pairs, corresponding to scenarios where $B$ has more elements equal to 1 than an identity matrix.
>
> ---
>
> **Q3.** In this last case, which seems to be the choice of the paper, however, the probability of being a positive pair is still attributed to the views, while it is simply B, the matrix sanctioning the connections. Therefore, it seems that wxx’ is either 1 or 0.
>
> **A3.** Please see **A1**.
>
> ---
>
> **Q4.** In the proof of Theorem 2.4, it seems that it is possible to choose a patch both from the natural image and a patch from some of its views.
>
> **A4.** In our modeling, we choose patches from the natural image. Although we use the statement “$p$ is a random patch from the view $x$”, this is only the result of the mapping process in step 3 of A2.

---

> ### Author Response · Authors · 2023-11-19
>
> # Response to Reviewer MfsU (2/4)
>
> **Q5.** Then, suppose that $\overline{p}$ is a patch from $\overline{x}$, and that $x$ is a random crop of $\overline{x}$ that does not include $\overline{p}$, and $p$ is a patch from $x$. Then $p$ and $\overline{p}$ could even be of different classes, as argued above. Therefore, what is the meaning of the sentence “$p$ is the corresponding patch of $\overline{p}$”?
>
> **A5.** The statement "$p$ is the corresponding patch of $\overline{p}$." refers to that $p$ and $\overline{p}$ belong to the same position in the original image (Please refer to A2 for the process of generating positive patch pairs). It is worth noting that, in the first paragraph of section 2.2, we assume that no patches will be cropped out during data augmentations on natural images, which aligns with the settings of some existing methods. This assumption ensures the clarity that $\overline{p}$ will not be cropped out; otherwise, it may result in no $p$ being generated.
>
> ---
>
> **Q6.** Furthermore, a clear definition of the encoding f for patches is not given, as it is provided only for views (see page 12 Proof of Theorem 2.3); however, in the proof of Theorem 2.4, $f(p)$ is used.
>
> **A6.** We apologize for the prior inconsistency in symbol usage within the paper. To differentiate symbols for GCL and DCL, we have introduced the subscript $I$ (Image-level) for GCL. Furthermore, we have adopted a consistent usage of the lowercase $f_I$ and $f$ to denote the network, and the symbol $F$ no longer appears. We have rectified these issues in the updated PDF (mainly in Sections 2.1, 2.2, and Appendix C). We appreciate your thorough review and constructive suggestions.
>
> ---
>
> **Q7.**  $F$ is used in many equations (e.g. in equation 1, it should be $f$) without being suitably introduced.
>
> **A7.** See A6.
>
> ---
>
> **Q8.** In Theorem 2.3, the function $f$ does not appear, though it appears in C Proof, page 12, and it is not said that f should be an embedding function, and here, too, $F$ remains purposeless.
>
> **A8.** See A6.
>
> ---
>
> **Q9.** Also, it seems that $\psi^y$ cannot be defined as $2\epsilon$ since the similarity between augmentations (including cropping) is different from the similarity between views as stated here, where patch/cropping and other augmentations are decoupled.
>
> **A9.** The term $\phi^{\hat{y}}$ characterizes the label difference between two augmented data. In the case of GCL, as in Haochen's proof (section C.2 of our paper), we have $\phi^{\hat{y}_I} \leq 2\epsilon$, where $\epsilon$ is defined in assumption 2.2 and is close to 0.
>
> However, in the case of DCL, $\phi^{\hat{y}}$ no longer exhibits a significant correlation with $\epsilon$. Instead, its value is notably influenced by the strategy for selecting positive pairs. For example, an aggressive strategy may result in matching patches that are not semantically consistent, thereby increasing the value of $\phi^{\hat{y}}$. To account for this influence, we introduce the notation $\alpha$ to represent this quantity and proceed with the subsequent discussion.
>
> ---
>
> **Q10.** The relation between $B$ as a circulant matrix and the general matrix must be discussed. I could not find an explanation in the appendix.
>
> **A10.** We observe that the circulant matrix characterizes the positive pair selecting strategy that matches spatially adjacent patches as positive pairs under circular padding. Consequently, we consider the circulant matrix as a particular instance of the matrix $B$ and carry out theoretical analysis. Additionally, to offer a more complete discussion, we conduct experimental analysis considering the scenario where $B$ takes the form of an arbitrary matrix. We have added explanations in the PDF (in the middle of page 5).
>
> ---
>
> **Q11.** In theorem 2.8, it seems that $h(l)$ is not used. Further, the sentence “the spatial distance of a pair will simultaneously increase, which means a larger labeling error rate $α$” should be made precise.
>
> **A11.** Certainly, the variable $h(l)$ is not further utilized in subsequent discussions. Its primary role is to illustrate that, in the context of the circulant matrix, the error bound $\mathcal{E}(f)$ demonstrates a negative correlation with $l$. This correlation reveals the connection between eigenvalues (which lack intuitive meaning) and the number of positive pairs (which carry intuitive meaning).
>
> For the second question, we have changed the statement to “…… the spatial distance of a pair will simultaneously increase. **Intuitively, as the spatial distance between patches grows, the probability of them being semantically close decreases.** This phenomenon results in a higher labeling error rate $\alpha$.” (On the top of page 6)

---

> ### Author Response · Authors · 2023-11-19
>
> # Response to Reviewer MfsU (3/4)
>
> **Q12.** D in the appendix comes from HaoChen et al. (2021).
>
> **A12.** We have chosen to relocate some of the introductory content from the preliminary section to the appendix for the sake of simplifying the main text, which is the role of Appendix D. The role of Appendix D is added in the PDF. (On the top of page 3)
>
> ---
>
> **Q13.** In Table 2, it probably needs to be noticed that for COCO instance segmentation, Mask R-CNN is used.
>
> **A13.** Thanks for your correction. In fact, we use Mask R-CNN (R50-FPN) for COCO tasks. We have already updated it in the PDF. (On the top of page 8)
>
> ---
>
> **Q14.** No sensible new theory is added. Furthermore, many statements remain unjustified. The overall writing quality needs improvement in terms of clarity and organization.
>
> **A14.** We note that [1] only establishes the theory of GCL methods (using image-level features) for image-level linear classification tasks. It is **not clear whether these image-level features could really generalize to dense prediction tasks like detection**. To fill this gap in theory, our work generalized [1] to analyze DCL methods and **establish formal generalization guarantees for dense prediction tasks for the first time**. Our analysis explicitly models the role of local information that is novel to the field and makes the following contributions:
>
> - **A mathematical formulation for DCL and dense prediction tasks.** Different from GCL, which aligns image-level representations, DCL aligns **local objects/patches** and achieves superior performance in local prediction tasks. However, it is unknown whether DCL methods also have guarantees for dense prediction tasks as in  GCL methods [1]. **Our work fills this gap by proposing a principled formulation for dense prediction tasks.** We regard each patch as a sample and establish a dense augmentation graph to model their relationship specified by different DCL methods in a unified framework. We believe this new formulation could lay the basis for future theoretical studies of dense prediction tasks.
> - **Analyzing the influence of local information through decoupling.** The uniqueness of DCL lies in the use of local information to establish positive pairs, while directly adapting [1] hinders it from being explicitly analyzed. To address this problem, we are **the first to decouple the effect of image-level similarity and patch-level similarity** through the Kronecker product. In this way, we can explicitly characterize the influence of local matching on downstream performance, leading to practical guidelines for the design of DCL methods.
> - **A new synthetic model for dense-labeled data that nicely verifies our theory**. We propose a new synthetic model to synthesize the dense-labeled data via the Gaussian Mixture Model, which is illustrated to be a good proxy for the real-world data generation process of images (Figure 3). Under this model, we explicitly calculate the labeling error $\alpha$ and graph eigenvalue $\mu_{k+1}$, and **the result aligns very well with our theory (Table 1)**.
> - **New locality-aware metrics for evaluating DCL methods**. Different from GCL methods that compare image representations [1], our designed evaluation metrics (Sec 4) include the local information into design, which makes it more suitable for predicting performance on dense prediction tasks, as we verified through real-world experiments (Table 3).
>
> To summarize, our theory indeed **takes the specialty of DCL methods and dense prediction tasks into consideration in every step: the formulation, the analysis, the verification, and the proposed metrics.** We believe that our analysis serves as a good starting point for studying the generalization properties of dense prediction tasks instead of focusing only on image-level tasks like [1].
>
> Lastly, as mentioned in other answers, we have fixed many typos to improve the clarity and organization of the paper.
>
> [1] HaoChen, J. Z., Wei, C., Gaidon, A., and Ma, T. Provable guarantees for self-supervised deep learning with spectral contrastive loss. In NeurIPS, 2021.
>
> ---
>
> **Q15.** The authors state that their theoretical framework can be extended to other approaches, but they only experiment on one, namely PixContrast.
>
> **A15.** Besides PixPro and PixContrast, in Appendix B, we conduct experiments on another DCL method, SoCo, which is also compatible with our framework as discussed.

---

> ### Author Response · Authors · 2023-11-19
>
> # Response to Reviewer MfsU (4/4)
>
> **Q16.** In the paper, eigenvalue distributions are discussed about graph matrices. How were these distributions simulated, and what significance do they hold for the study?
>
> **A16.** We assume that the reviewer refers to the simulation experiments on the eigenvalues of a general matrix. As mentioned in Appendix A, we compute the eigenvalue distributions of the normalized form of random symmetric boolean matrices with varying numbers of elements equal to 1. We start with an identity matrix. In each iteration, we randomly change a 0 element to 1. In the Figure 3, the x-axis denotes the number of iterations, and the y-axis denotes the eigenvalues of the normalized form (mentioned in section 2.1) of the matrix. All the intersections of the curves in the figure and the vertical line induced by the function $x=i$ form the eigenvalue distribution on the iteration $i$.
>
> This experiment shows that even without the assumption of a circulant matrix, the eigenvalue $\mu_{j}$ is negatively correlated with the number of positive pairs, which serves as an important support to our claim of the trade-off.
>
> ---
>
> **Q17.** Why is the eigenvalue µ_(k+1) of B a good surrogate to compute error? (See Training and Evaluation section).
>
> **A17.** The objective of the simulation experiment is to illustrate the trade-off proposed by our theory. Despite the noted difficulty in simulating the image-level positive-pair graph, which is stated in the paper, it remains unchanged in our setting. In this context, according to Lemma 2.6, the eigenvalues of $\bar{A}$ are positively correlated with the eigenvalues of the strategy matrix $B$, which is $\mu_{(k+1)}$. Consequently, we have employed $\mu_{(k+1)}$ as a surrogate to compute the error.
>
> ---
>
> **Q18.** In the explanation on page 7, last paragraph, what is meant in the sentence “positions in the original image are smaller than a given threshold”?
>
> **A18.** We apologize for the confusion in the statement. We have modified it into “… when **the distance of** their corresponding positions in the original image is smaller than a given threshold.”(on the bottom of page 7). Thanks again for your detailed review.
>
> ---
>
> **Q19.** Most notably, why strictly follow HaoChen et al. (2021), while the authors have written that it is hard to extend their approach to object detection and segmentation.
>
> **A19.** As mentioned in the introduction (the second paragraph in the updated PDF), we only claimed that extending existing theoretical results from GCL to DCL offers limited additional insights into DCL. In section 2.2, we presented a straightforward extension of Haochen's construction to DCL, and showed that this extension did not allow for the decoupling of image-level and patch-level similarity. This limits the discussion on pixel-level supervision, which is the essence of DCL.
>
> However, the uniqueness of DCL lies in the utilization of local information for establishing positive pairs. Consequently, we introduced a decoupling mechanism for image-level and patch-level similarity through the Kronecker product. This innovative approach enables the explicit characterization of the influence of local matching on downstream performance. As a result, we can provide practical guidelines for the design of DCL methods, offering a unique perspective beyond the extension of existing GCL methods.
>
> ---
>
> Thank you again for your review. We have made many updates to the theory and expressions of our work, and address each of your concerns above. We respectfully suggest that you could re-evaluate our work based on these updated results. We are very happy to address your remaining concerns about our work during the discussion stage.

---

> ### Comment · Reviewer_MfsU · 2023-11-19
> **Thank you**
>
> Thank you so much for your reply.
> You are writing:
>
>   " 1.  Selection of patches from a natural image.
>     2.  Application of data augmentations, including cropping, on the natural image.
>     3.  During augmentation, the selected patches are mapped to corresponding patches on each augmented view.
>     4.  Utilization of a positive pair selection strategy to match these patches into positive pairs.
> ...
> It is worth noting that, in the first paragraph of section 2.2, we assume that no patches will be cropped out during data augmentations on natural images, which aligns with the settings of some existing methods. This assumption ensures the clarity that \hat{p}
> will not be cropped out; otherwise, it may result in no being generated."
>
> But in fact, this is exactly what I have written:
>
>
> 1. if cropping is not allowed and patches are chosen on the natural image,
> 2. then \mathcal A, namely the probability, given a natural image \overline{x},
> 3. that the augmentation x and x’ form a positive pair, makes the problem trivial.
> 4. In fact, on the image \overline{x}, whatever patch is chosen, the augmentation can stretch them, change colour, blur or rotate them; they will always be of the same class if they come from the same region. Therefore, B is trivial: only patches from different locations can differ.

---

> ### Author Response · Authors · 2023-11-20
> **Replying to "Thank you"**
>
> We apologize for the ambiguity in our rebuttal expression. The phrase “be cropped out” actually means the patched “is removed from the cropping process” but not “is generated by cropping”. Actually, in the original PDF (4th line of section 2.2), our expression is “we assume that image-level data augmentation preserves all patches in the image for the patch-level matching task.” To add clarity to the expression, we have modified it into “we assume that image-level data augmentation (including RandomResizedCrop) preserves all patches in the image for the patch-level matching task.”
>
> About the four points the reviewer listed:
>
> 1. (Point 1) As mentioned above, cropping **is** allowed and patches are chosen on the natural image.
>
> 2. (Point 2 & 3) We still need to emphasize that $\mathcal{A}$ and $w_{xx'}$ are non-trivial **regardless of** whether cropping is allowed and whether the classes of augmentation views are identical. The reviewer might consider that $w_{xx'}=1$ when the classes of $x$ and $x’$ are identical and otherwise $w_{xx'}=0$. In fact, in its definition $w_{xx'} = E_{\overline{x} \in P_{\overline{\mathcal{X}}}}(A(x|\overline{x}) A(x'|\overline{x}))$, no class is appeared. Besides, $\mathcal{A}(x|\overline{x})$ refers to the probability of generating $x$ from $\overline{x}$, which is also irrelevant to the class information, but only how many augmented views are generated.
>
> 3. (Point 4) Since $B$ is defined by the strategy for selecting positive pairs, it has no relationship with the ground-truth class of the patches. While the reviewer correctly notes that only patches from different locations can differ in ground-truth classes, resulting in non-trivial $B$, it is essential to highlight that in the strategies for selecting positive pairs, we indeed allow for the matching of patches from different locations to form positive pairs (regardless of their classes), leading to a non-identical $B$. This aligns precisely with the approach of many methods, such as PixPro.
>
> The key point is that the ground-truth classes of images or patches do not impact the problem setting and mathematical modeling since this is under the framework of self-supervised learning. They solely influence the downstream generalization error. As mentioned in sections 2.2 and 2.3, the generalization error is positively correlated with the labeling error $\alpha$, which is influenced by the ground-truth classes.

---

> > ### Comment · Reviewer_MfsU · 2023-11-20
> > **simple problem**
> >
> > As the title says, the problem is simple.
> >
> > Let us go back to HaoChen and colleagues:
> > "Our approach is based on the central concept of population augmentation graph, denoted by
> > G(X, w), where the vertex set is all augmentation data X and w denotes the edge weights defined
> > below. For any two augmented data x, x'\in X, define the weight w_{xx'} as the marginal probability of
> > generating the pair x and x' from a random natural data \hat{x} ~ PX:
> >
> > w_{xx'} := E_{x~PX}[A(x|\hat{x})A(x'|\hat{x})]      (*)
> >
> > Therefore, the weights sum to 1 because the total probability mass is 1: \sum_{x,x'}\in X w_{x,x'} = 1."
> >
> > This was HaoChen and colleagues.
> >
> > Now, to build the normalized adjacency matrix you want the probability to give you the edge connection. In your case, the problem is quite simple, as far as you do as follows:
> > 1. Suppose the number of images is M
> > 2. Transform an image in N images. For example, take the transformer, transform an image in N patches, patches are indexed, and you know precisely the correspondence between patches, then instead of the set of images X you have the set of patches X_{set of patches}, whose cardinality is N*M
> >
> > 3. Having the precise correspondences treat each patch as HowChen and colleagues treat images: resize, random crop, etc. then you can:
> > "For any two augmented data x, x'\in X_{set of patches}, define the weight w_{xx'} as the marginal probability of
> > generating the pair x and x' from a random natural data \hat{x} ~ PX_{set of patches}.
> > 3. You do not even need to give a special size to the Adjacency matrix since you can rename NM as K.
> >
> > This would solve the problem, without any need to add further B to ( * ), since instead of having M images you'll simply have N*M of them.  But then it is interesting to analyse how good should the patch dimension be, and so on, because, for example, cropping could destroy any possible interpretation of the image. Thus a strategy for augmentation should be given.
> >
> > I think that if you correct all the indicated errors about denotations etc., and if you make the mapping from images to patches quite simple and straightforward as I suggested, the paper might be appreciable.

---

> > > ### Author Response · Authors · 2023-11-20
> > > **Replying to "simple problem"**
> > >
> > > Thanks for your detailed reply! Now we understand your point on simplifying the presentation logic. Indeed, by letting $w_{pp’}$ be the probability of generating two patches $p$, $p’$ from a random natural data $\overline{p}$, i.e., $w_{pp'} = E_{\overline{p} \in P_{\overline{\mathcal{X}}_{\text{P}}}}(A(p|\overline{p}) A(p'|\overline{p}))$, we can give a direct construction of the generation of positive patch pairs.  **Following this suggestion, we have adopted this modeling in section 2.2 in the newest PDF.**
> > >
> > > Meanwhile, we still want to note that this patch-level augmentation graph $A$ is only a straightforward adaptation of Haochen et al., and does not offer much insight into positive pair selection. Therefore, we think that it would be helpful if **we could decompose $A$ into an image-level adjacency matrix $A_I$ and strategy matrix $B$,** and analyze their contributions to the downstream error separately.
> > >
> > > As for why this decoupling makes sense, we now understand your concern that random cropping from the raw input $\bar x$ into two views $x,x’$ may generate images with mismatching patches, which does not permit a universal $B$. We find that in practice, DCL methods usually address this problem by firstly finding the common area of $x$ and $x’$, and only performing dense alignment within this area, e.g., DUPR, PixPro. In the common area, the two views have the same patches, so **these methods indeed use a fixed alignment rule (i.e., a universal $B$) among patches**. Likewise, if we regard the image-level augmentation as a composition of the cropping operator and the taking-overlap operator, the generated views can still have aligned patches. As a result, we can perform the decomposition $A=A_I\otimes B$ with a universal $B$.
> > >
> > > Hope this clarification could address your concerns. We have added this new explanation to Sec 2.3 to explain the rationality of decoupling. Please let us if there is more to clarify. Besides, we’ve fixed many of the indicated errors about denotations during the previous rebuttal. Please inform us if there are any remaining issues that need to be revised.

---

> > > > ### Comment · Reviewer_MfsU · 2023-11-21
> > > > **OK thank you**
> > > >
> > > > Thank you for your answer.
> > > > In fact, my suggestion, wanted to show exactly what you said: the problem of collecting patches could be "a straightforward adaptation of Haochen et al.".
> > > > Though now you need something more, since you may have a huge amount of meaningless positive pairs that are neither classes nor elementary components of a precise class, maybe your B can solve this, since you know that a specific bunch of patches come all from a distinct class.

---

> ### Author Response · Authors · 2023-11-21
> **Replying to "OK thank you"**
>
> Thanks for improving the score! Indeed, as you pointed out, there might be some class-irrelevant patches (so-called, “meaningless”) in an image, such as patches from the sky, the streets, etc. According to our theory, aligning such patches may not contribute to understanding the centric object.
>
> Nevertheless, we note that dense contrastive learning is often applied to dense prediction tasks where every pixel has a label, such as, image segmentation. For such tasks, we need to understand the whole scene, so the alignment of these patches is still helpful. This perspective may explain why dense contrastive learning often outperforms global contrastive learning on these tasks.
>
> Hope this explanation could address your concerns. Please let us if there is more to clarify.

---

### Official Review · Reviewer_vGDf · 2023-10-31

**Soundness:** 3 good
**Presentation:** 3 good
**Contribution:** 3 good
**Rating:** 6
**Confidence:** 2

**Summary:**

The paper proposes the first theoretical framework to model and analyze dense contrastive learning (DCL) methods. This is done by constructing a patch-level positive pair graph and using concepts from spectral graph theory. By decomposing the adjacency matrix of the patch-level graph using the Kronecker product, the framework is able to decouple the image-level and patch-level supervision. Theoretical and experimental analysis shows there is a trade-off between the quantity and correctness of positive pairs selected in DCL. Matching more positive pairs reduces eigenvalues but increases labeling error rate. Two unsupervised metrics are introduced - Patch Confusion Rate (PCR) and patch-level contrastive loss - to guide positive pair selection in DCL without requiring ground truth labels. Overall, the paper provides the first theoretical understanding of DCL methods through the analysis of the patch-level positive pair graph and makes both theoretical and practical contributions.

**Strengths:**

1. This work first proposes a theoretical framework to model and analyze dense contrastive learning (DCL) methods.
2. Theoretical analysis is clear, and the experimental results can support the claims.
3. Two novel metrics are proposed: PCR and patch-level contrastive loss.

**Weaknesses:**

1. This paper does not show enough results to support the superiority of the proposed metrics. The experiment cannot support these metrics can be used to determine positive pair selection.
2. Experiment on a real dataset is needed to validate the application of proposed theory and metrics to improve DCL models.

**Questions:**

The authors mentioned that the proposed metrics can determine positive pair selection strategies. The authors reckon smaller PCR and lower loss reveals a better model, however, the results in Table 3 cannot prove it. The authors need explain more for the case 5×5.

---

> ### Author Response · Authors · 2023-11-19
>
> # Response to Reviewer vGDf (1/2)
>
> We thank Reviewer vGDF for appreciating our theory, the nice alignment between the theory and experiments, and the novelty of our proposed metrics. Below, we address your concerns about the experiments of our work.
>
> ---
>
> **Q1.** This paper does not show enough results to support the superiority of the proposed metrics. The experiment cannot support these metrics that can be used to determine positive pair selection.
>
> **A1.** Our proposed metrics not only illustrate a discernible trade-off resulting from our theoretical findings but also can be used to infer better model performance. As stated in the Q3 of the reviewer, “smaller PCR and lower loss reveal a better model”. Our experiments align with this principle. For example, the loss of PixPro is much lower than SimplePixPro-All, while their PCR is similar, indicating that PixPro is better than SimplePixPro-All. As another example, the PCR of PixPro is much smaller than SimplePixPro-1x1, but their loss is similar, indicating that PixPro is better than SimplePixPro-1x1.
>
> We are not certain about the specific source of confusion for the reviewer. It's possible that a point of confusion lies in the interpretation of the "loss" metric, which refers to its raw value rather than its absolute value. For instance, -3.58 (PixPro) is considered smaller than -1.44 (SimplePixPro-All). If there are particular aspects causing confusion, please provide additional details, and we will do our best to address and clarify those points.
>
> ---
>
> **Q2.** Experiment on a real dataset is needed to validate the application of the proposed theory and metrics to improve DCL models.
>
> **A2.** In Table 3, we have already validated that the application of the proposed theory and metrics can be used to improve DCL methods. For instance, when taking SimplePixPro-1x1 or SimplePixPro-All as the baseline, PixPro exhibits a superior strategy for selecting positive pairs for performance improvement. This is verified by our proposed metrics that both the PCR and loss of PixPro are lower. This serves as evidence that our proposed metrics can indeed aid in selecting a better positive pair selecting strategy, ultimately resulting in improved model performance.
>
> Besides, existing methods also contribute to validating our theory's insights on real datasets. Specifically, our theoretical analysis actually shows that the downstream generalization of DCL methods is determined by two factors: 1) the labeling error between positive patch pairs and 2) the connectivity between patches in the dataset. In fact, existing methods are working on these aspects. First, to reduce the labeling error, as done in similarity-based positive selection methods like PixPro, DenseCL [1], SelfPatch [2], and SetSim[3], they pick semantically similar patch pairs according to the estimated feature similarity. Moreover, DUPR [4] and SoCo [5] achieve this by considering the exact matching of positive pairs (i.e., $B = I$). Second, to increase data connectivity, as done in PixPro and SelfPatch, they adopt a feature aggregation module. The empirical success of these methods on real datasets helps verify our theory’s insights, and our theory can serve as a principled explanation for these modern DCL methods. We are also considering new solutions besides these existing ones, but considering the fact that this paper is mostly theory-oriented, we leave this part for future work.
>
> [1] Wang, X., Zhang, R., Shen, C., Kong, T., and Li, L. Dense contrastive learning for self-supervised visual pre-training. In *CVPR*, 2021.
>
> [2] Yun, Sukmin, et al. "Patch-level representation learning for self-supervised vision transformers." *In CVPR*. 2022.
>
> [3] Wang, Z., Li, Q., Zhang, G., Wan, P., Zheng, W., Wang, N., Gong, M., and Liu, T. Exploring set similarity for dense self-supervised representation learning. In *CVPR*, 2022
>
> [4] Ding, J., Xie, E., Xu, H., Jiang, C., Li, Z., Luo, P., and Xia, G.-S. Deeply unsupervised patch re-identification for pre-training object detectors. *IEEE Transactions on Pattern Analysis and Machine Intelligence*, 2022.
>
> [5] Wei, F., Gao, Y., Wu, Z., Hu, H., and Lin, S. Aligning pre-training for detection via object-level contrastive learning. In *NeurIPS*, 2021.

---

> > ### Comment · Reviewer_vGDf · 2023-11-21
> >
> > Thank you for your response.
> >
> > A1: I reckon the experiment in Table 3 is not convincing enough. First, it lacks sufficient data point to prove the conclusion with high statistical confidence. Therefore, my suggestion is to show the performance comparison with enough models. The comparing models can be median checkpoints, because the purpose of this experiment is to generate models with diverse abilities. Second, SPECTRAL PIXCON. 1x1 is the best model in the table. However, the PCR is the highest. The present experimental results did not show a trend of a performance increasement as reducing PCR.

---

> > > ### Author Response · Authors · 2023-11-22
> > >
> > > Thanks for your comments. We evaluate the performance (AP) on PASCAL VOC object detection, as well as the PCR and loss of PixPro and SimplePixPro using the intermediate results from 10 epochs.
> > >
> > > | Models | Neighbors | VOC AP | PCR($\downarrow$) | Loss($\downarrow$) |
> > > | --- | --- | --- | --- | --- |
> > > | Simple PixPro | $1\times1$ | 46.1 | 0.67 | -3.49 |
> > > |  | $3\times3$ | 45.4 | 0.37 | -1.43 |
> > > |  | $5\times5$ | 42.3 | 0.16 | -0.80 |
> > > |  | All | 44.6 | 0.62 | -1.19 |
> > > | PixPro | - | 49.8 | 0.55 | -2.18 |
> > >
> > > In this table, we can still observe the trade-off between PCR (connectivity) and loss (labeling error). Furthermore, we can also identify a superior model based on the proposed loss. For instance, compared with SimplePixPro-All, both PixPro and SimplePixPro-3x3 have lower PCR and loss, indicating that PixPro/SimplePixPro-3x3 outperforms SimplePixPro-All.
> > >
> > > Furthermore, you mentioned that among Spectral PixContrast models, the 1x1 model is the best but has the highest PCR. However, it also has the lowest loss (labeling error), which mitigates the impact of its high PCR and ultimately leads it to being the best model. We hope that our original and additional experiments can address your concerns.

---

> ### Author Response · Authors · 2023-11-19
>
> # Response to Reviewer vGDf (2/2)
>
> **Q3.** The authors mentioned that the proposed metrics can determine positive pair selection strategies. The authors reckon smaller PCR and lower loss reveals a better model, however, the results in Table 3 cannot prove it. The authors need to explain more for the case 5×5.
>
> **A3.**
> In fact, the case 5x5 aligns well with our theory. Among PixPro or PixContrast methods, the 5x5 cases both have the lowest PCR but the highest loss (note that for loss we refer to the raw value, which means -0.55 is larger than -0.78 or other loss values). This is in line with the trade-off proposed by our theory. On the other hand, it is not conclusive to assert that their model performance is either the best or the worst, according to the principle that “smaller PCR and lower loss reveal a better model”.
>
> ---
>
> Thanks for your careful reading and detailed review. Hope our explanations can address your concern. Please let us know if you have additional questions.

---

> ### Author Response · Authors · 2023-11-23
>
> Dear Reviewer vGDf,
>
> We have prepared a response to address your additional questions. Would you please take a look and let us know whether you find it satisfactory?
>
> We note that Reviewer MfsU has appreciated our response and raised the score. We also respectfully suggest that you could re-evaluate our work with the updated results and explanations.
>
> Thanks! Have a great day!
>
> Authors

---

### Meta-Review · Area_Chair_zwZx · 2023-12-17

**Metareview:**

The paper examines dense contrastive learning through the lens of spectral clustering on the positive-pair graph.

Though all three reviewers give "marginal accept" rating, the weaknesses outlined by the reviewers and their comments in the subsequent discussion with the authors reveal serious shortcomings of the paper.  The AC does not believe the content of these assessments is consistent with an accept recommendation.

Reviewers vGDf points to insufficient experiments, stating "This paper does not show enough results to support the superiority of the proposed metrics" and is not swayed in subsequent discussion.  Reviewer MfsU raises issues with the assumptions in the method regarding positive pair selection, which, even after a discussion thread with the authors, is not clearly resolved.  Reviewer MfsU also points to concerns in presentation and experiments and states "No sensible new theory is added", "many statements remain unjustified" and "overall writing quality needs improvement in terms of clarity and organization."

**Justification For Why Not Higher Score:**

A clearer presentation of the theoretical claims, along with more substantial experimental evidence is needed to address concerns raised by reviewers.

**Justification For Why Not Lower Score:**

N/A

---

### Decision · Program_Chairs · 2024-01-16

Reject